# Inflammaging in Multidrug-Resistant Sepsis of Geriatric ICU Patients and Healthcare Challenges

**DOI:** 10.3390/geriatrics9020045

**Published:** 2024-04-03

**Authors:** Nishitha R. Kumar, Tejashree A. Balraj, Kusuma K. Shivashankar, Tejaswini C. Jayaram, Akila Prashant

**Affiliations:** 1Department of Biochemistry, JSS Medical College and Hospital, JSS Academy of Higher Education & Research, Mysuru 570015, India; nishitharkumar@jssuni.edu.in (N.R.K.); kusumaks@jssuni.edu.in (K.K.S.); 2Department of Microbiology, JSS Medical College and Hospital, JSS Academy of Higher Education & Research, Mysuru 570015, India; tejashreea@jssuni.edu.in; 3Department of Geriatrics, JSS Medical College and Hospital, JSS Academy of Higher Education & Research, Mysuru 570015, India; tejaswinicj@jssuni.edu.in; 4Department of Medical Genetics, JSS Medical College and Hospital, JSS Academy of Higher Education & Research, Mysuru 570015, India

**Keywords:** multidrug-resistant sepsis, intensive care unit, geriatric, immunosuppression, inflammation

## Abstract

Multidrug-resistant sepsis (MDR) is a pressing concern in intensive care unit (ICU) settings, specifically among geriatric patients who experience age-related immune system changes and comorbidities. The aim of this review is to explore the clinical impact of MDR sepsis in geriatric ICU patients and shed light on healthcare challenges associated with its management. We conducted a comprehensive literature search using the National Center for Biotechnology Information (NCBI) and Google Scholar search engines. Our search incorporated keywords such as “multidrug-resistant sepsis” OR “MDR sepsis”, “geriatric ICU patients” OR “elderly ICU patients”, and “complications”, “healthcare burdens”, “diagnostic challenges”, and “healthcare challenges” associated with MDR sepsis in “ICU patients” and “geriatric/elderly ICU patients”. This review explores the specific risk factors contributing to MDR sepsis, the complexities of diagnostic challenges, and the healthcare burden faced by elderly ICU patients. Notably, the elderly population bears a higher burden of MDR sepsis (57.5%), influenced by various factors, including comorbidities, immunosuppression, age-related immune changes, and resource-limited ICU settings. Furthermore, sepsis imposes a significant economic burden on healthcare systems, with annual costs exceeding $27 billion in the USA. These findings underscore the urgency of addressing MDR sepsis in geriatric ICU patients and the need for tailored interventions to improve outcomes and reduce healthcare costs.

## 1. Introduction

Geriatric patients are more prone to developing infections, particularly multidrug-resistant (MDR) sepsis, due to age-related changes in their immune system, comorbidities, and medication [1]. This colonization is further exacerbated by an alarmingly increasing frequency (75%) of MDR infections in older patients admitted to the intensive care units (ICU) of hospitals [2]. Patients afflicted with MDR infections face a greater likelihood of being transferred to other healthcare facilities, extended hospital stays, elevated healthcare costs, and heightened all-cause in-hospital mortality [3]. In particular, ICU patients, particularly those who are immunocompromised due to organ transplantation, previous antibiotic exposure, or the presence of central venous catheters, are at an increased risk of acquiring MDR infections [4]. The rapid proliferation of MDR pathogens has raised concerns regarding the adequacy of empirical antibiotic therapy, which is of critical concern in hospital ICUs [5]. In the USA, the annual incidence of antibiotic-resistant bacteria in critically ill patients is linked to over 700,000 hospital-acquired infections (HAIs) [6].

As the threat of MDR sepsis continues to grow, geriatric patients in the ICU are at an increased risk, with potentially severe consequences. The diminishing effectiveness of antibiotics could exponentially increase the risk associated with surgical and medical procedures as well as immunosuppressive treatments, including cancer chemotherapy. Furthermore, the economic implications could be as catastrophic as the 2008–2009 global financial crisis [7]. Therefore, the aim of this review is to identify the clinical impact and provide valuable insights regarding healthcare challenges associated with managing MDR sepsis in geriatric ICU patients.

## 2. Intricacies and Clinical Impact of MDR Sepsis in Geriatric ICU Patients

In a study conducted in Australia over a period of 15 years, it was observed that among 4137 ICU patients over the age of 80 years with sepsis, chronic cardiovascular disease (9.5%), chronic respiratory illness (7.3%), and type II diabetes mellitus (4.8%) were the most prevalent comorbidities [8]. These comorbidities have underlying mechanisms that contribute to an increased risk of infection. For instance, diabetes mellitus has been associated with a heightened infection risk due to factors such as peripheral neuropathy, poor vasculature, and delayed pathogen clearance by neutrophils [9]. The older adults, ≥65 years, are thirteen times more likely to be hospitalized with sepsis than individuals <65 years. Nearly 2% of them are also more prone to readmission to the hospital ICU due to sepsis within 3 months as compared to those with non-sepsis hospitalization [10]. Similarly, immune deficits contributing to infections were reported in sepsis patients with chronic renal disease and chronic liver failure [11]. Another study indicated that poor health status is indirectly related to infection, hospitalization, and ICU admission among older individuals [12]. Additionally, recent studies revealed that severe sepsis patients are more susceptible to developing dementia after hospitalization [13]. However, it is important to note that only a limited number of studies have focused on ICU-acquired infections in geriatric patients, and their findings have been a subject of controversy. Most of these studies were conducted as single-center investigations, primarily focusing on specific types of infections [14,15]. One notable exception is the “Extended Prevalence of Infection in Intensive Care II (EPIC II)” study, which analyzed age-related infection patterns and outcomes in 1265 ICUs across 75 countries. This comprehensive study revealed that patients aged 85 years and older exhibited a higher incidence of gastrointestinal infections, a greater prevalence of Gram-negative bacteria, and higher mortality rates when compared to their younger counterparts [16]. Recently, there has been a growing emphasis on investigating age-specific alterations in organ dysfunction along with MDR infections. The heart comprises cardiomyocytes, cardiac fibroblasts, and macrophages as its primary defense [17]. Over time, the heart experiences changes characterized by several distinct features, such as cardiomyocytes gradually enlarging, cardiac fibrosis emerging, and inflammation becoming evident, collectively defining the senescing process within the heart [18,19]. The inflammation in senescent bone marrow impairs its function. This can lead to changes in the number and function of immune cells, which differentiate into various types [19]. Immune cell senescence is characterized by a decline in performing immune functions and an increase in the release of inflammatory factors, as illustrated in Figure 1.

## 3. Epidemiology and Healthcare Costs of Sepsis in Geriatric Patients

Among geriatric trauma patients, the incidence of sepsis is documented at 2.1% [20], with notably worse outcomes among those affected. Septic patients face a notably elevated risk of mortality, with reported rates of 6%, 15%, and 34% for sepsis, severe sepsis, and septic shock [21], respectively. The costs associated with managing sepsis range from approximately $16,000 to $25,000, contingent upon the severity level. In the United States, the total national hospital expenditure for sepsis care amounted to $16.7 billion in 1995, with a significant portion attributed to patients aged 65 and above [20]. The economic implications of sepsis management are substantial, varying depending on severity levels and whether sepsis was present upon admission or developed during hospitalization [21,22]. These findings emphasize the crucial need for promptly identifying and vigorously managing sepsis in geriatric patients to enhance outcomes and alleviate the economic strain on healthcare systems.

Between 2022 and 2050, the population of Americans aged ≥65 years is expected to surge from 58 million to 82 million, marking a 47% increase. Moreover, the proportion of individuals in the geriatric age group relative to the total population is forecasted to elevate from 17% to 23%. According to senior citizen reforms in India, by 2050, India is expected to accommodate 319 million older adults, comprising 20% of its entire population. This demographic shift is poised to further burden India’s healthcare system, which is already struggling with excessive demands. Additionally, projections suggest that nearly half (45%) of India’s disease burden will be shouldered by geriatric patients by 2030. This escalation is anticipated as age groups prone to severe conditions are set to constitute a larger segment of the population. Since the introduction of the initial consensus definition of sepsis (Sepsis-1) in 1991, the global prevalence and incidence of sepsis and septic shock have steadily increased. In 2017, an estimated 49 million cases of sepsis were confirmed worldwide, resulting in approximately 11 million sepsis-related mortalities [23]. For instance, a Chinese study conducted between 2017 and 2019 reported an annual increase in hospitalized sepsis cases, rising from 328.25 to 421.85 cases per 100,000 individuals. In the context of ICUs, sepsis affects a substantial proportion of patients with significant regional differences [24]. Notably, regions such as Oceania, sub-Saharan Africa, and various Asian regions, including South, Southeast, and East Asia, exhibit higher prevalence rates. Furthermore, sepsis does not discriminate by age or gender, affecting individuals across the demographic spectrum. However, there are considerable differences in the burden of illness, with susceptible groups, particularly the elderly, individuals with chronic illnesses, and those with compromised immune systems, being more profoundly affected [25]. A study spanning from 2017 to 2019 found a sepsis incidence rate of 57.5% among individuals aged 65 years and older [26].

The economic impact of sepsis on healthcare systems is substantial. In the USA, the annual healthcare costs associated with sepsis were $20 billion in 2011 [27] and increasing to $24 billion in 2013–2014 [28], eventually reaching $27 billion in 2019. Notably, sepsis is the most expensive hospitalized illness in the USA, accounting for approximately $38 billion in annual healthcare expenditure [29]. In India, the estimated sepsis cost per patient was $55 in 2005 [30] while a 2008 study proposed a projected estimate of $53 million for the Indian healthcare system in 2012 [31]. Before the COVID-19 pandemic, Ontario and Canada spent an estimated $1.3 billion annually on sepsis-related healthcare expenses [29]. In a nationwide Japanese study, sepsis cases surged from 67,318 in 2010 to 233,825 in 2017, resulting in an adjusted annual gross medical cost increase from $3.04 billion to $4.38 billion [32]. These escalating healthcare costs can be attributed to prolonged hospital stays, expensive medications, and, regrettably, restricted access to treatment for sepsis patients, contributing to an alarming number of misdiagnosed sepsis-related fatalities [33].

## 4. Factors Contributing to the Rise of MDR in Sepsis in Geriatric ICU Patients

In the clinical management of sepsis, physicians often strive to provide effective empirical antimicrobial treatment for hospitalized patients, sometimes resorting to prescribing antibiotics without precise diagnostic confirmation. This practice, while intended to save lives, may cause the unfortunate expense of potentially prescribing unnecessary antibiotics. Such excessive antibiotic use has been linked to the emergence and proliferation of MDR bacteria. Beyond clinical practices, several other factors contribute to the rise of MDR sepsis. A higher incidence of MDR sepsis can be attributed to multiple patient-specific factors. These include older age with uncommon clinical presentations necessitating frequent low-potent antibiotic usage, the presence of comorbidities, states of immunosuppression or the overuse of immunosuppressive drugs, chemotherapy regimens for cancer patients, and living in countries with lower- and middle-income economies marked by deprived healthcare infrastructure and limited accessibility to healthcare facilities [34]. Polypragmasy, which involves the concurrent use of five or more drugs, is another significant factor contributing to the development of MDR bacteria in sepsis cases. Polypragmasy is often associated with the natural process of aging, which, due to simultaneous biological and pathological changes, elevates the risk of comorbidities and the necessity for multiple medications concurrently [35]. As bacterial populations have evolved and proliferated, the efficacy of empirical antimicrobial therapy has diminished, contributing to the rise in multidrug-resistant (MDR) sepsis. Antibiotics have become less effective as microorganisms have adapted, evolved, and continued replicating even in their presence, as shown in Figure 2.

Consequently, most bacteria have developed acquired resistance to one or more antibiotics. In contrast, many bacteria can adapt by altering their antibiotic-targeting sites, leading to antibiotic resistance [36,37]. Similarly, self-medication practices by individuals, often involving nonspecific treatments for a range of ailments, can inadvertently contribute to the development of resistance among opportunistic pathogens [38]. The escalation in MDR sepsis causing hospital readmissions is attributed to many factors, such as inadequate infection control, the overuse of a broad spectrum of antibiotics, not finishing the course of prescribed antibiotics, or using antibiotics when not needed, which can contribute to the emergence of MDR bacteria [37]. Immunocompromised patients, such as those with chronic illnesses or receiving treatments such as chemotherapy, are more susceptible to MDR infections [39]. Delayed diagnosis or the inappropriate treatment of severe sepsis can lead to the persistence and spread of MDR bacteria [40]. Antibiotics failing to locate their bacteriostatic and bactericidal targets signify non-specific treatment, potentially accelerating the emergence of multidrug-resistant sepsis [41]. The mode of action of the pharmacokinetics (pK) and pharmacodynamics (pD) of antibiotics is shown in Figure 3.

## 5. Common Pathogens Involved in MDR Sepsis

Many microbes can be the causative agents of sepsis, including bacteria, viruses, fungi, and even parasites. Among these, bacteria are the most prevalent aetiological pathogens associated with sepsis and sepsis-related comorbidities. In geriatric patients, respiratory tract and genitourinary infections are the most common source of sepsis [42,43]. This population is more prone to infections caused by MDR pathogens due to immune senescence, long-term exposure to institutionalization, and the excessive and early use of broad-spectrum antimicrobials, which may lead to the emergence of MDR in pathogens [42,44]. Common bacterial pathogens implicated in sepsis include: *Streptococcus pneumoniae*, a leading cause of community-acquired pneumonia, which can progress to sepsis, particularly in vulnerable populations; *Staphylococcus aureus*, as both methicillin-sensitive and methicillin-resistant strains of *S. aureus* can cause severe infections, leading to sepsis; *Escherichia coli*, a common Gram-negative bacterium, responsible for urinary tract infections that may result in sepsis; *Hemophilus influenzae*, which causes invasive infections such as pneumonia and meningitis, with sepsis as a possible complication; *Salmonella* spp., which can cause severe gastrointestinal infections that may progress to sepsis; and *Neisseria meningitidis*, a known cause of meningococcal meningitis, which can rapidly lead to septicemia and sepsis [45]. Some researchers revealed a higher incidence of sepsis-causing MDR pathogens, methicillin-resistant *S. aureus*, and vancomycin-resistant *Enterococci* in geriatric patients. The incidence of extended-spectrum β-lactamase-producing *Klebsiella* species was also found to be higher in geriatric patients (>65 years) compared to those <14 years [42,43]. Similarly, fungal infections account for a significant portion of sepsis cases, particularly in immunocompromised or critically ill patients. Specifically, Candida species are the prominent cause of fungal sepsis, accounting for 5% of all sepsis cases. Invasive Candida infections are linked with a higher risk of sepsis-associated mortality. Several studies have emphasized that inadequate treatment and delays in administering appropriate antifungal medications are linked to higher mortality rates in patients with candidemia or septic shock [46]. Additionally, seasonal or periodic influenza, dengue viruses, and highly contagious pathogens of community health significance can also induce sepsis and septic shock. Notable examples include swine and avian influenza viruses, severe acute respiratory syndrome-related (SARS) coronavirus, Middle East respiratory syndrome-related (MERS) coronavirus, and, most recently, Ebola and yellow fever viruses [47].

## 6. Mechanism of Inflammation on MDR Sepsis and Aging in Critical Care

MDR sepsis poses formidable challenges within ICUs, significantly impacting patient discharge outcomes and straining healthcare resources. The complex nature of MDR microorganisms overshadows antimicrobial treatment approaches, often resulting in treatment failures and prolonged hospitalizations. These resistant microorganisms raise concerns about possible horizontal transmission within ICUs, highlighting the vital need for consistent infection prevention and control policies. Resource-restricted ICUs face unique challenges, including limited access to essential equipment, laboratory assistance, and qualified physicians and nursing teams and poor hand hygiene. Therefore, the guidelines for sepsis management in areas with limited resources, such as those formulated by the Global Intensive Care Working Group of the European Society of Intensive Care Medicine (ESICM) [48], differ from recommendations issued by the Surviving Sepsis Campaign (SSC), which are primarily based on data from high-income economies [49].

Moreover, multiple infections, including ventilator-associated pneumonia (VAP) and hospital-acquired pneumonia (HAP), represent prevalent HAIs in the ICUs, accounting for a significant proportion of antibiotics administered in critical care settings. Despite ongoing efforts to improve the early detection and therapy of sepsis and septic shock, morbidity and mortality remain high, especially in MDR sepsis patients [50]. In the past decade, the widespread use of β-lactam antibiotics for the treatment of VAP and HAP in ICUs has revealed neurotoxic symptoms in approximately 10–15% of ICU patients. Similarly, combining these antibiotics with nephrotoxic drugs, such as vancomycin, has been associated with an increased incidence of renal complications in sepsis patients [51].

Sepsis results from an uncontrolled immune reaction to infections, thereby disturbing a balance of inflammatory responses to maintain homeostasis. When a pathogen invades the human body, the immune system identifies its molecular components (pathogen-associated molecular patterns, PAMPs) through specific receptors called pattern recognition receptors (PRRs). This activation induces the production of inflammatory cytokines (causing leukocyte activation) and activates complement and coagulation systems, which initiate a detrimental cycle that ultimately progresses to sepsis. Immunosuppression occurs as extensive apoptosis, which causes immune cell depletion. Similarly, organs initiate inflammation and activate compensatory mechanisms to maintain homeostasis. These responses are critical to coping with infection, injury, stress, and other challenges to ensure proper functioning and survival. MDR pathogens induce acute respiratory distress syndrome (ARDS), a severe and potentially life-threatening condition characterized by the frequent onset of extensive inflammation in the lungs. Moreover, severe MDR bacterial infections cause cytokine storms, lung injuries, secondary infections, and ventilator-associated pneumonia. MDR sepsis causes systemic vasodilation and reduced blood flow, which places a massive burden on the heart, leading to compromised cardiac function and possible myocardial damage. MDR sepsis, together with immune suppression, elevates the risk of complications [52].

Geriatric patients are particularly susceptible to sepsis due to immunosuppression, pre-existing comorbidities, diminished aging-associated physiological reserves, sarcopenia, malnutrition, and polypragmasy [53]. Immune senescence and inflammaging are two critical processes that make geriatric patients more vulnerable to sepsis [54,55]. Immune senescence is a consistent decline in immunity, particularly the function of T cells, while inflammaging is persistent low-grade inflammation. Immune senescence and inflammaging are interconnected and initiate a cycle that increases an individual’s susceptibility [55,56,57,58]. Similarly, the interaction of the immune system with other body systems, such as endocrine or neural systems, establishes a link between declining immunity and conditions such as frailty, sarcopenia, and malnutrition [54,55]. Lower metabolism and reduced insulation further compromise the immunity of geriatric patients, making them prone to infections and illnesses. Geriatric syndromes resulting from multi-system impairment result from various age-related changes, comorbidities, and environmental impacts, which all together significantly affect the quality of life and increase patient susceptibility to infection. Frailty is more common with aging; it affects 25% of geriatric patients over 65 and over 50% of patients over 80, affecting around 40% of geriatric ICU patients, leading to significantly high morbidity and mortality [59,60,61,62,63]. Similarly, the prevalence of sarcopenia is between 11% and 50% for geriatric patients of 80 years and above [64]. Aging interrupts muscle balance and induces mechanisms, including anabolic resistance, inflammation, oxidative stress, and mitochondrial dysfunction. Anabolic resistance reduces muscle response to stimuli, thereby causing muscle wasting and diminished protein synthesis. Immobilization in hospitalized geriatric patients causes a daily 0.5% and 0.3–4.2% muscle mass reduction and a strength decline, respectively, which impacts the body’s functional status and the quality of life of these patients [65]. Whereas sepsis further worsens the sarcopenia, sarcopenia is connected to multiple pathophysiological processes, which exacerbate inflammation, muscle wasting, and mitochondrial dysfunction [66,67], thereby increasing the risk of mortality in critical conditions such as sepsis.

For instance, antibiotic-resistant *Klebsiella* species release PAMPs, including lipopolysaccharides (LPS), which interact with PRRs on immune cells [68,69]. This interaction initiates intracellular signaling cascades, activating transcription factors such as NF-κB and AP-1, leading to the production and release of pro-inflammatory cytokines such as interleukin-6 (IL-6) and tumor necrosis factor-alpha (TNF-α) [70,71]. The early cytokine response serves as an alarm signal, attracting more immune cells to the site of infection, enhancing phagocytosis, and initiating adaptive immune responses [72]. However, dysregulated cytokine production can lead to a cytokine storm, as illustrated in Figure 4, exacerbating tissue damage and organ dysfunction, characteristic of severe sepsis and septic shock in geriatric ICU patients.

The interaction between sepsis and aging involves a complex interplay of acute and chronic inflammatory responses, which can have profound implications for the clinical course, management, and outcomes of sepsis in older adults. Table 1 illustrates the interaction between the specificities of sepsis and aging at the acute and chronic inflammatory programming responses mentioned below.

## 7. Diagnostic Challenges and Innovations of MDR Sepsis in Geriatric Patients

The term “geriatric onset” is often used to differentiate between conditions that primarily affect older adults and those that can occur at any age [73]. Understanding the age at which a condition such as neurodegenerative diseases, such as Alzheimer’s or Parkinson’s disease, cardiovascular diseases, osteoporosis, certain cancers, and various other age-related health issues typically begins can be important for diagnosis strategies [74,75]. Coexisting chronic conditions in geriatric patients can complicate the clinical diagnosis results and make it challenging to distinguish sepsis from above underlying conditions [76]. Similarly, a timely diagnosis and treatment are critical in improving clinical outcomes and reducing sepsis-related mortality. Conventionally, the diagnosis of sepsis involves a serum analysis and molecular diagnostic techniques such as polymerase chain reaction (PCR) and isothermal amplification. However, these approaches are labour-intensive, resource-demanding, and time-consuming and require skilled personnel [77]. In contrast, innovative diagnostic methods have emerged as promising alternatives. Surface-enhanced Raman spectroscopy (SERs) [78] and matrix-assisted laser desorption ionization time-of-flight mass spectrometry (MALDI-TOF-MS) [79] have demonstrated the potential for the rapid detection of sepsis-related pathogens. These techniques offer advantages such as speed and accuracy. Moreover, automated bioinformatics tools, such as whole-genome sequencing (WGS) [80], have enabled the tracking of antibiotic resistance genes. The advent of next-generation sequencing (NGS) has further facilitated large-scale WGS, making it more accessible and cost-effective. Similarly, the microarray technique can effectively identify microbes through surface-immobilized DNA and RNA probes [81]. This method reduces sample and reagent consumption, thereby lowering costs and enabling the accurate segregation of microbial species, even at the strain level.

Furthermore, point-of-care sensors have emerged as valuable tools for timely sepsis diagnosis and intervention. These sensors rapidly compile patient health data, increase healthcare coverage, and enhance service efficiency while simultaneously reducing healthcare costs [82]. POCT-based devices can identify pathogens, cell-surface proteins, and plasma proteins. When combined with extensive data analytics [83], POCT can assist in stratifying sepsis, even at the patient’s bedside, and rapidly detect patients who may benefit from supplementary therapy. Moreover, POCT can aid in antibiotic selection by evaluating protein biomarkers (including IL-6, IL-10, PCT and CRP, and TNF-ɑ) associated with acute sepsis and septic shock in ICU patients, providing estimates of the probability of all-cause mortality within 28 days [84]. Moreover, syndromic testing on geriatric patients for molecular diagnostic techniques can simultaneously detect multiple pathogens and their resistance genes, providing clinicians with innovative information for deciding antibiotic therapy.

## 8. Clinical Management of MDR Sepsis in Geriatric ICU Patients

The choice of antimicrobials for MDR sepsis in geriatric ICU patients is a complex process that considers multiple factors. These factors include the patient’s medical history, comorbidities, immune status, clinical presentation, suspected source of infection, presence of invasive devices, morphological data of microorganisms, and regional prevalence and resistance patterns of pathogenic bacteria [85]. For most sepsis patients without septic shock, empirical broad-spectrum treatment with one or multiple antimicrobial drugs is the recommended approach to target a wide range of potential pathogenic microorganisms [86,87]. However, patients with sepsis and septic shock may require initial combination therapy with two antimicrobials from two different classes based on identified pathogens and local or regional antibiotic susceptibility patterns. As sepsis with aging involves a complex pathophysiology, including early inflammatory and later immunosuppressive responses, particularly in geriatric patients, combining pharmacological treatments such as thymosin ɑ1 (Tɑ1) and ulinastatin (UTI) or interleukin-7 with anti-PD-1 monoclonal antibodies holds a promise as a treatment strategy for sepsis [88].

Moreover, tailoring treatment for geriatric ICU patients requires a nuanced approach, focusing on individualized therapy guided by clinical management strategies. Consideration of patient-specific conditions is paramount, especially in cases of MDR sepsis where comorbidities play a crucial role in treatment outcomes [89]. For instance, conditions such as diabetes can compromise immune function, while renal or hepatic dysfunction may affect drug metabolism, and immunosuppression heightens susceptibility to infections [90]. Tailoring treatment strategies must factor in these influences to optimize outcomes and minimize complications. Antibiotic selection, a key aspect, hinges on various factors, including the site of infection, suspected pathogens, antimicrobial susceptibility profiles, and patient-specific risk factors [91,92]. Immunocompromised patients may necessitate broad-spectrum antibiotics or combination therapy to combat resistant pathogens effectively and mitigate treatment failure risks [92,93]. Dosing optimization is imperative to achieve therapeutic concentrations while mitigating toxicity risks, particularly in patients with impaired renal or hepatic function [93]. Vigilantly monitoring for adverse effects and the proactive management of potential complications are essential components of individualized therapy for MDR sepsis, ensuring optimal treatment response and patient safety [94]. Therefore, tailoring treatment to these patient-specific characteristics can yield favorable results instead of applying a uniform approach to all patients [95]. Similarly, personalized immunomodulatory treatment, tailored to the patient’s immune profile, may be more effective for treating sepsis [96]. Furthermore, supportive care plays a vital role in the management of sepsis and includes hemodynamic support to maintain tissue perfusion, fluid resuscitation, and vasopressors, if needed. Similarly, mechanical ventilation techniques, such as low tidal volume ventilation and prone positioning, have demonstrated benefits in patients with sepsis-induced acute respiratory distress syndrome [97,98].

## 9. Preventive Measures and Infection Control Strategies Impacting MDR Sepsis in Aging

Recently, the idea and significance of the Antimicrobial Stewardship Program (ASP) has gained significant recognition as a crucial tool in the battle against antibiotic resistance. The ASP is commonly characterized as a cohesive set of initiatives designed to promote the appropriate and responsible use of antimicrobial drugs [99]. The misuse and abuse of antimicrobials are among the most pressing global public health challenges, leading to the emergence of antibiotic resistance. Therefore, the ASP represents an organized and systematic program for promoting the appropriate usage of antimicrobials, enhancing patient outcomes, and limiting the transmission of MDR infections [100]. Understanding the routes and trends of environmental contamination in the transmission of MDR pathogens can guide healthcare personnel in designing more effective prevention techniques. Environmental cultures, including swab testing and the monitoring of water and air samples, provide invaluable data on the prevalence and persistence of MDR bacteria. These insights help establish connections between environmental pollution and pathogen acquisition [101].

Similarly, educating healthcare professionals about the principles of the ASP strengthens preventative interventions by encouraging the prudent use of antibiotics, thereby reducing the emergence and spread of MDR bacteria [102]. Despite the challenges associated with implementation, it is clear that ASPs have a favorable impact on infection management in ICU settings [103]. Furthermore, integrating innovative technologies, such as artificial intelligence (AI) and machine learning, can enhance the effectiveness of ASPs [104]. Cultivating a stewardship culture and implementing a one health strategy, which acknowledges the interrelatedness of environmental, animal, and human health, are essential elements in the fight against MDR bacteria in ICUs [105]. Vaccination represents another important preventive measure, often administered before bacterial growth or spread following the initial infection and before various tissues and organs are affected. This significantly lowers the probability of mutations that confer resistance from arising and spreading [106].

Particularly, aging exacerbates the rate of sarcopenia and frailty post-sepsis, leading to accelerated muscle loss and increased frailty in older adults. This results from sepsis-induced muscle wasting, prolonged immobility, and pre-existing age-related muscle decline [107]. Aging widens the spectrum of severity in sarcopenia and frailty, influencing outcomes in older populations. Age-specific interventions, including early rehabilitation, nutritional support, and comprehensive geriatric assessment, are vital for optimizing outcomes, especially for individuals over 65 who face higher risks of severe and prolonged muscle loss and frailty [108]. Sarcopenia, driven by anabolic resistance, reduced IGF-1 signaling, and inflammation, contributes to muscle loss, while frailty represents increased vulnerability due to aging-associated decline, impacting mortality and morbidity rates significantly among older ICU patients [109]. Prioritizing tailored interventions addresses the critical needs of aging populations grappling with post-sepsis sarcopenia and frailty [108,109].

## 10. Future Directions in Research and Therapeutics

The future directions in research and therapeutics for multidrug-resistant (MDR) sepsis in geriatric patients encompass a multifaceted approach aimed at addressing the unique challenges posed by this population. Key areas of focus include the development of new antibiotics or the repurposing of existing drugs to combat MDR bacteria, a critical step in overcoming the growing issue of antibiotic resistance [94]. Additionally, there is a push towards investigating novel biomarkers that can improve the diagnosis of sepsis and risk stratification in elderly patients, particularly those with frailty, to ensure timely and appropriate interventions. Exploring alternative treatment strategies, such as immunomodulatory therapies, is also pivotal in enhancing the immune response of older adults, who may have diminished immune function due to age-related changes. Personalized care is another significant area, with an emphasis on developing individualized treatment plans that consider factors such as frailty, comorbidities, and patient values, acknowledging the complexity and heterogeneity of the geriatric population [107]. Improving data collection and standardization is crucial for better understanding the epidemiology of sepsis in older adults, which can guide research, policy, and clinical practice. Lastly, there is a need for strategies that improve early diagnosis and intervention in older adults, who are at higher risk for delayed diagnosis and poor outcomes, to mitigate the significant morbidity and mortality associated with sepsis in this age group [107,108].

Similarly, the field of sepsis care has witnessed remarkable transformations in therapeutic approaches, including significant advancements in immunomodulatory drugs and targeted therapies aimed at mitigating excessive inflammatory responses [110,111]. The escalating challenge of antimicrobial resistance has prompted the development of novel techniques to combat antibiotic resistance effectively [112]. These strategies encompass supplementing antibiotics with adjunct therapies, optimizing supportive care, targeting bacterial virulence factors, and addressing host response factors to improve antibiotic effectiveness. For example, hemadsorption methods, such as polymyxin B adsorption, show potential in filtering out endotoxins and mitigating the detrimental effects of septic shock [113]. Researchers have also identified various antibiotic alternatives, including phage therapy, which has shown promise in preclinical and clinical experiments [114]. However, it is crucial to emphasize that these innovative techniques require rigorous validation and integration into comprehensive care paradigms.

Moreover, recent advancements in immune medicine have prompted renewed research into sepsis immune therapy. Precision medicine, guided by genomics, proteomics, metabolomics, and point-of-care technologies, offers individualized immune-based treatments, including monoclonal and polyclonal antibodies and immunomodulators tailored to the specific patient profile [115]. The overarching goal of this approach is to finely tune immunological responses to minimize the damage associated with sepsis [116].

## 11. Conclusions

Combating MDR sepsis in geriatric ICU patients requires a multifaceted approach to conclude a rapid and accurate diagnosis, prompt antimicrobial therapy, and comprehensive organ support. MDR sepsis in geriatric ICU patients presents significant challenges. Sepsis, as a multifaceted immune dysfunction, underscores the critical importance of a timely diagnosis. Each hour’s delay in a sepsis diagnosis can profoundly impact a patient’s recovery and increase hospital-acquired mortality among sepsis patients. Therefore, the imperative lies in the early and adequate administration of antimicrobials, preferably within the first hour of diagnosis, coupled with essential organ support. Public health organizations, such as the World Health Organization, engage in global collaborations with stakeholders to enhance the treatment of sepsis and fortify infection prevention control measures. Technological advancements have ushered in a pivotal role for POCT in the bedside detection of sepsis. Additionally, interventions such as nanoparticles and immune-based therapies, bolstered by precision medicine, are made familiar with paramount considerations for healthcare providers across diverse specialties, including physicians, pharmacists, and microbiologists. Furthermore, cultivating public awareness regarding antimicrobial resistance and addressing its multifaceted mystifying challenges, particularly in geriatric patients, remain paramount. Effective treatment strategies and minimizing adverse effects hinge on well-informed public and collaborative efforts on a global scale.

## Figures and Tables

**Figure 1 geriatrics-09-00045-f001:**
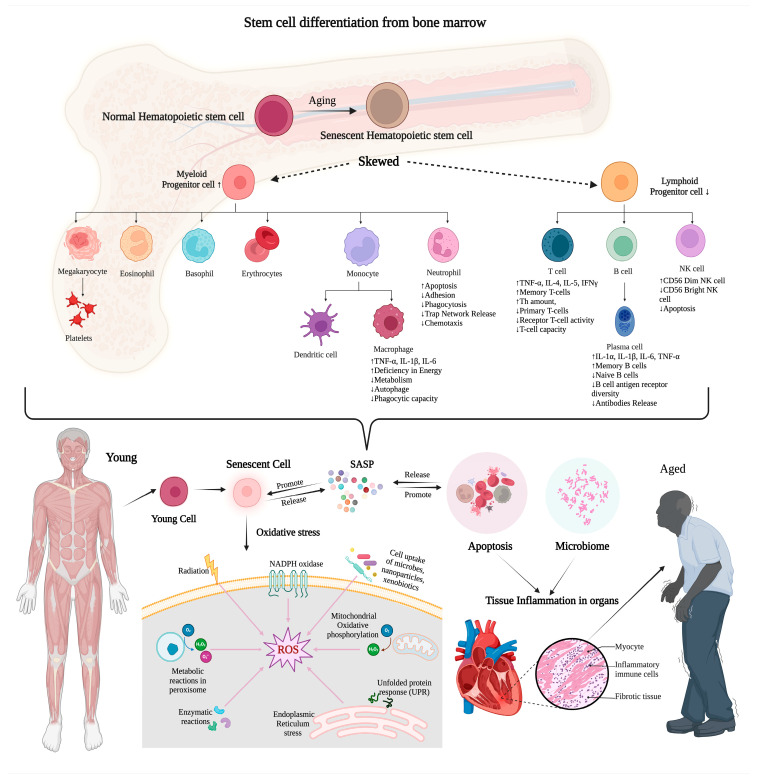
Inflammaging in senescence and immune dysfunction. Senescent cells are characterized by dysfunction and the acquisition of a senescence-associated secretory phenotype (SASP) accumulating throughout the body. Immune cells are crucial for senescent cell clearance and are also affected by SASP, leading to immunosenescence. This decline in immune function compromises the body’s ability to combat infections and diseases, increasing susceptibility to illnesses. Additionally, senescent cell accumulation triggers inflammation within organs, contributing to organ damage and the heightened risk of age-related diseases. Positive feedback loops perpetuate inflammation and organ damage, exacerbating the risk of aging-related diseases. The up arrow indicates an increased level/activity, the down arrow indicates a decreased level/activity.

**Figure 2 geriatrics-09-00045-f002:**
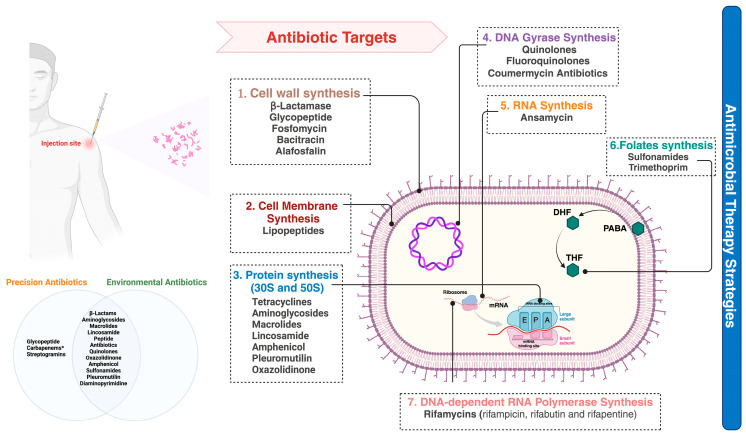
Empirical antimicrobial therapy strategies in the proliferation of bacteria contributing to a rise in MDR sepsis. Antibiotics have increasingly disorientated their efficacy as microorganisms have evolved, grown, and reproduced even in their presence. A bacterial infection is the starting point for treatment with human and environmental factors. There are classes of antibiotics shown in the Venn diagram. (*) Carbapenems are a sub-class of β-lactam antibiotics mainly used as precision medicine; carbapenem, streptogramins, and glycopeptides are only used for human treatments. All classes of antimicrobials are widely used to treat environmental factors. Self-medication, often involving broad-spectrum antibiotics for various ailments, further exacerbates the problem by providing selective pressure for the survival of resistant strains. (1) Inhibition in the growth of bacteria by targeting the bacterial cell wall; (2) the cell membrane inhibitor synthesis; (3) the latter is a process performed by ribosomes, nucleoprotein complexes, which consist of a small and large subunit (30S and 50S in bacteria). (4) Antibiotics can inhibit DNA gyrase, an enzyme which modifies the DNA conformation, playing a role in replication and transcription. (5) Antibiotics can inhibit RNA synthesis by binding to the beta-subunit of bacterial RNA polymerase. (6) Antibiotics act as antimetabolites by inhibiting the folate metabolism (and consequently the DNA inhibitor synthesis) in a pathway involving paraaminobenzoic acid (PABA) and two precursors of folic acid, dihydrofolic acid (DHF), and tetrahydrofolic acid (THF). (7) Antibiotics can inhibit bacterial DNA-dependent RNA polymerase essential for the transcription of messenger RNA in bacteria.

**Figure 3 geriatrics-09-00045-f003:**
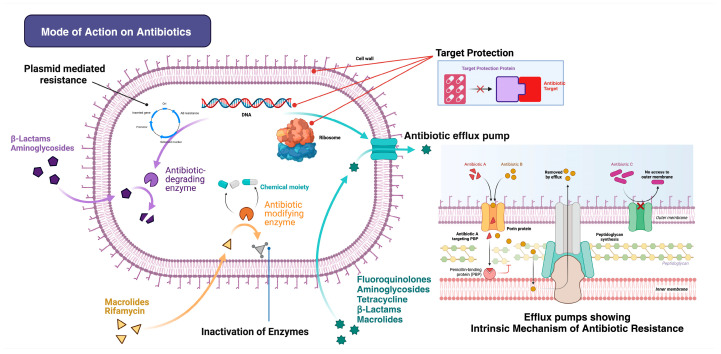
The action of antibiotics providing resistant bacteria leads to hospital readmission. Action Mechanisms 1: The efficacy of antibiotics is diminishing due to reduced penetration into bacterial cells. Bacteria acquire or develop resistance (target protection, target bypass, target site modification) to antibiotics by reducing the antibiotic intracellular concentration because of their low penetration into the bacteria (aminoglycosides, β-lactams, aminoglycosides, quinolones). Action Mechanisms 2: Bacteria can circumvent the effects of antibiotics by actively expelling them from their internal structures. Bacterial resistance to antibiotics is facilitated by efflux pumps, which reduce the intracellular concentration (inactivation of the enzyme, degrading the antibiotic molecule), thereby propelling the development of resistance (e.g., aminoglycosides β-lactams, fluoroquinolones macrolides, quinolones). Action Mechanism 3: The roles of porins and efflux pumps in bacteria can be graphically demonstrated, depicting how porins facilitate the entry of substances and efflux pumps enable the exit of antibiotics. Porins, which are protein formations in the membrane, and efflux pump transport proteins play key roles in moving molecules in and out of cells. There are five main categories of efflux transporters: (1) resistance–nodulation–cell division family, (2) small multidrug resistance family, (3) major facilitator superfamily, (4) multidrug and toxic compound extrusion family, (5) ATP-binding cassette superfamily. As demonstrated in the provided figure, molecules are propelled out of cells by efflux pumps as H+ or Na+ ions are ushered in. For members of the ABC family, ATP is the impetus for their function. This energy-carrying molecule is broken down into ADP and inorganic phosphate to power their operation.

**Figure 4 geriatrics-09-00045-f004:**
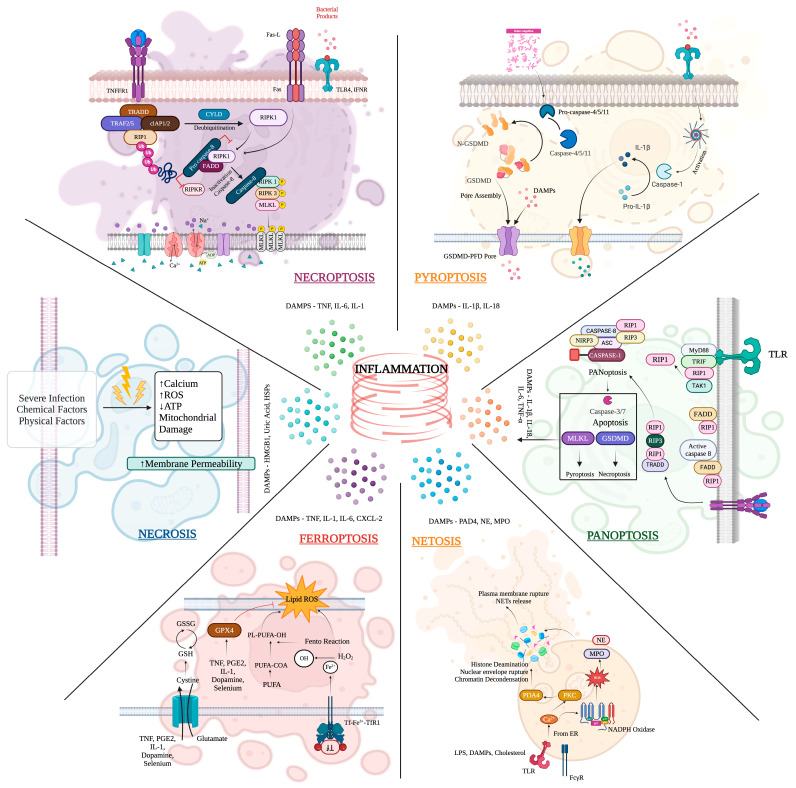
An overview of six molecular patterns associated with inflammatory cell death. Inflammatory cell death, including apoptosis, necroptosis, pyroptosis, ferroptosis, autophagy-dependent cell death, and NETosis, plays a crucial role in various physiological and pathological processes, including immune responses, tissue repair, and disease pathogenesis. Understanding the distinct molecular mechanisms and signalling pathways of cell death is essential for elucidating their contributions to inflammation-related aging and developing targeted therapeutic strategies.

**Table 1 geriatrics-09-00045-t001:** Interaction between the specificities of sepsis and aging in acute and chronic inflammatory responses.

Aspect	Sepsis	Aging	Interaction
Acute inflammatory response	Triggers dysregulated systemic inflammation	Chronic low-grade inflammation	Exaggerated acute response due to pre-existing chronic inflammation
	Excessive release of pro-inflammatory cytokines (e.g., IL-6, TNF-α)	Increased production of pro-inflammatory cytokines	Amplified tissue damage and organ dysfunction
Chronic inflammatory response	Leads to persistent inflammation post-recovery	Presence of high inflammation	Chronic inflammation post-sepsis may exacerbate inflammatory response
	Prolonged immune dysregulation	Immunosenescence (decline in immune function)	Increased susceptibility to recurrent infections
	Associated with long-term complications	Contributes to chronic diseases	Higher risk of functional decline and recurrent infections

## Data Availability

Not Applicable.

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
