# Peer review of "Inflammaging in Multidrug-Resistant Sepsis of Geriatric ICU Patients and Healthcare Challenges"

_geriatrics, 2024, doi:10.3390/geriatrics9020045_

Round 1
Reviewer 1 Report
Comments and Suggestions for Authors
I applaud the hard work on a difficult subject in need of enlightenment. I am concerned the forest of sepsis per se, obscures the trees. In other words the narrative settled too much on sepsis without enough specific coupling to geriatrics. Aging and sepsis is the issue, not reviewing sepsis and not the figures without integrating aging/ geriatrics.
How do the specificities of sepsis interact with the specificities of aging at the acute and chronic inflammatory programming responses of each?
Sepsis review dominates, leaving geriatrics plus sepsis not adequately integrated. Needed is a skillful editor of content with context.
The title is inappropriate: Aging and Sepsis might be better.
How do MDR microbes differentially impact aging associated sepsis. How does the rapid sepsis microbial resistance component of sepsis specifically impact aging and vice versus? And how does the sepsis disease/sickness response with profoundly immunosuppressive immunometabolic paralysis. How do the two impacting immunity and metabolism interact? What are the missing pieces? What specific insights can geriatrics offer. What about chronic sepsis and geriatrics.
i suggest a content and context editor cam help reduce content by 49-50 pecent, leaving enriched context.
As the aging population grows and grows, this paper can become seminal.
“This review aims to explore the clinical impact of MDR sepsis in geriatric ICU patients and shed light on healthcare challenges associated with its management.”The title is a misnomer, with review—-important in intent——straying back and forth between sepsis and geriatrics. Precise connections are missing, and would have important implications. Publications and narrative that start with geriatrics and then connect to aging specificities are needed.
For example, how does antibiotic-resistant Klebsiella species specifically invade to propel early cytokine attempt and does it specifically create the sickness or disease response. Although such two are not mutually distinct they are two critical components of the sepsis landscape. Is this landscape distinct to aging and if so, how?
The tile is misleading with limited information about MDR bacteria. I suggest a title such as Aging and Sepsis.
I suggest focus on geriatrician information that has distinct messages and not just a dialogue review of sepsis, of which many are readily available.
I suggest that each section include aging or geriatrics in the title and the action better inform the subtypes.
Examples: Do MDR microbes have specific impact on the aging hosts the better inform geriatrocs, do they selectively increase carrier states before after sepsis?
What is the impact of aging on the rate and spectrum of post sepsis sarcopenia and frailty and is are age specific—not just reiterating known features of the syndrome.
Geriatrics should have the front position in each section.
I suggest shortening the narrative by geriatric focus, using a context and content editor
I suggest the Figures change or be replace by visual information that includes aging connections distinct from just listing germs and pathways.
2) A skilled context and content editor could CLARIFY INFORMATION and REDUCE CONTENT BY 50%
Comments on the Quality of English Language
There is excessive rambling and disconnection of context with excessive content. As a results the wordy and misplaced context wanders from a focus on specifically coupling aging with distinct features of geriatrjcs and with distictict features of sepsis and MDR microbes.
Author Response
Respected Reviewer,
We have made the changes accordingly to the reviewer’s suggestions and comments in our manuscript and changed the title to “Inflammaging in Multidrug-Resistant Sepsis of Geriatrics ICU patients and Healthcare Challenges”.
We have addressed the reviewer’s suggestions and comments below.
- We addressed the reason for the aging and sepsis as an issue integrating to senescence molecular, cellular, and tissue dysfunction and provided in the Figure1. (Page 3)
- We have provided a Table 1 relating to specificities of sepsis interacting with aging at acute and chronic inflammatory responses. (Page 10)
- We addressed the comments MDR sepsis, together with immune suppression, elevates the risk of complications in aging with microbial resistance. (Page 6)
- We have revealed the common MDR pathogens to be higher in geriatrics patients. (Page 6 & 7)
- We have illustrated an overview of six molecular patterns associated with inflammatory cell death in Figure 4. (Page 9)
- We explained the tailoring clinical management strategies of MDR sepsis associated to geriatrics ICU patients. (Page 11)
- We have given the reason for antibiotic resistant Klebsiella species specifically invade to propel early cytokine to cause immune response. (Page 8)
- We have addressed the impact of aging and rate and post sepsis sarcopenia featuring preventive measures and infection control of MDR. (Page 11 & 12)
- We have addressed the future directions in research and therapeutics of MDR sepsis in geriatrics patients. (Page 12)
- We have addressed intricacies and clinical impact of MDR sepsis in geriatric patients (Page 2)
- As reviewer’s suggestions we have addressed the epidemiology and healthcare cost of sepsis in geriatrics patients (Page 3)
- We have compelled geriatrics in front position in all the section.
- All the changes are high lightened as yellow in the manuscript and few of the minor comments are addressed inside the manuscript.
Thank you for your time and consideration.
Reviewer 2 Report
Comments and Suggestions for Authors
This is a very general review, without accent on geriatric population with sepsis. There is no strong relation between title of the chapters and content. There are more powerful information in medical literature about this subject that could be used as reference.

Comments on the Quality of English Language
minor editing of English language required
Author Response
Respected Reviewer,
We have made the changes accordingly to the reviewer’s suggestions and comments in our manuscript and changed the title to “Inflammaging in Multidrug-Resistant Sepsis of Geriatrics ICU patients and Healthcare Challenges”.
We have addressed the reviewer’s suggestions and comments below.
- We addressed the reason for the aging and sepsis as an issue integrating to senescence molecular, cellular, and tissue dysfunction and provided in the Figure1. (Page 3)
- We have provided a Table 1 relating to specificities of sepsis interacting with aging at acute and chronic inflammatory responses. (Page 10)
- We addressed the comments MDR sepsis, together with immune suppression, elevates the risk of complications in aging with microbial resistance. (Page 6)
- We have revealed the common MDR pathogens to be higher in geriatrics patients. (Page 6 & 7)
- We have illustrated an overview of six molecular patterns associated with inflammatory cell death in Figure 4. (Page 9)
- We explained the tailoring clinical management strategies of MDR sepsis associated to geriatrics ICU patients. (Page 11)
- We have given the reason for antibiotic resistant Klebsiella species specifically invade to propel early cytokine to cause immune response. (Page 8)
- We have addressed the impact of aging and rate and post sepsis sarcopenia featuring preventive measures and infection control of MDR. (Page 11 & 12)
- We have addressed the future directions in research and therapeutics of MDR sepsis in geriatric patients. (Page 12)
- We have addressed intricacies and clinical impact of MDR sepsis in geriatric patients (Page 2)
- As reviewer’s suggestions we have addressed the epidemiology and healthcare cost of sepsis in geriatrics patients (Page 3)
- We have compelled geriatrics in front position in all the section.
- All the changes are high lightened as yellow in the manuscript and few of the minor comments are addressed inside the manuscript.
Thank you for your time and consideration.
Round 2
Reviewer 1 Report
Comments and Suggestions for Authors
I accept this second submission entering your publication route.
Comments on the Quality of English Language
none